# United on Sunday: The effects of secular rituals on social bonding and affect

**Sarah J. Charles** [1]*, **Valerie van Mulukom**[1], **Jennifer E. Brown**[1], **Fraser Watts**[2], **Robin I. M. Dunbar**[3], **Miguel Farias**[1]

1 Brain, Belief, and Behaviour Lab, Centre for Trust, Peace and Social Relations, Coventry University, Coventry, United Kingdom, 2 International Society for Science and Religion, Cambridge, United Kingdom, 3 Department of Experimental Psychology, University of Oxford, Oxford, United Kingdom

* charle42@uni.coventry.ac.uk

**Data Availability Statement:** All data and supplementary materials can be found on the related OSF project: 10.17605/OSF.IO/NPDZM.

**Funding:** YES MF, RIMD and FW were recipients of a Templeton Religion Trust grant (Grant number

## Abstract

Religious rituals are associated with health benefits, potentially produced via social bonding. It is unknown whether secular rituals similarly increase social bonding. We conducted a field study with individuals who celebrate secular rituals at Sunday Assemblies and compared them with participants attending Christian rituals. We assessed levels of social bonding and affect before and after the rituals. Results showed the increase in social bonding taking place in secular rituals is comparable to religious rituals. We also found that both sets of rituals increased positive affect and decreased negative affect, and that the change in positive affect predicted the change in social bonding observed. Together these results suggest that secular rituals might play a similar role to religious ones in fostering feelings of social connection and boosting positive affect.

## Introduction

There is a substantial body of research demonstrating beneficial health and wellbeing effects that stem from the attendance of religious rituals (for an overview, see [1]). The current evidence suggests that the benefits of religious ritual attendance include improved wellbeing [2], as well as protection against all-cause mortality [3–6], depression [7–9], suicidality [10] and immune dysfunction [11]. Moreover, VanderWeele [1] notes that many of these positive effects appear to be best maintained when ritual attendance is at least once per month. Much of this literature was conducted in western, democratic nations, and within Christian settings, though there are some notable exceptions to this [12–14].

A classical assumption of sociological and anthropological sciences is that that one of the primary functions of religion is to promote group solidarity (e.g. [15]). Dunbar [16] has suggested that religious rituals developed as a mechanism to help form and maintain social bonds in groups of humans in particularly effective. A wealth of evidence supports this theory, showing that many of the behaviours that are incorporated into religious rituals lead to feelings of social bonding, such as joint attention [17–19], shared goals [20], synchronised movement [21–23], music making [24–26], eating [27], and moderate alcohol consumption [28, 29]. Accordingly, religious rituals that incorporate a multitude of these behaviours should foster bonds efficiently [16].

0153). The Trust's website can be found here: https://templetonreligiontrust.org/ The funders did not play a role in the design of the study nor in data collection, analysis, decision to publish or preparation of the manuscript.

**Competing interests:** The authors have declared that no competing interests exist.

Recently, Charles and colleagues [30] provided the first evidence that religious rituals (Christian and Afro-Brazilian) directly increase feelings of social bonding with other attendees. They collected data in over 20 religious rituals in the UK and Brazil, assessing self-reported levels of social bonding before and then after the rituals. Their results showed that taking part in these rituals significantly increased social bonding towards the group. They also reported that increased social bonding was significantly predicted by an increase in positive affect, a decrease in negative affect, a purported increase in endorphins (as reflected by increases in pain threshold) and by feelings of connection to God during the ritual, but not by age, gender, or religiosity.

This not only demonstrated that rituals lead to social bonding, but also provided support for the hypothesis that emotional state–and more specifically positive affect–is related to feelings of social connectedness [19, 31, 32]. This 'Broaden and Build' hypothesis [17, 19, 32, 33] suggests that positive affect leads to wellbeing improvements via the broadening of social relationships which then builds an individual's social support network. The broadening is said to occur as positive affect allows for an increasing scope of attention and behaviours to facilitate bonding with others. This hypothesis also serves as the basis for the proposal that the link between religion and wellbeing stems from changes in positive affect [34, 35], which in turn can lead to improved social bonding, and wellbeing [36]. Therefore, the mechanism by which religious rituals lead to social bonding could be by incorporating behaviours that lead to positive affect, which then contributes to social bonding.

There is increasing evidence for the link between social bonding and health: having strong social bonds has been shown to lead to improved health outcomes compared to those who lack them [37, 38], in the form of reduced mortality [39, 40] and lowering levels of depression, suicidality, and immune dysfunction [41, 42]. These health benefits overlap with those that religious rituals are purported to provide [1]. Consequently, the Broaden and Build hypothesis links religious ritual to wellbeing via positive emotions' role in broadening social bonding [34, 35].

Charles and colleagues [30] also showed that the feeling of connection to God during the ritual positively predicted changes in social bonding, albeit to a lower level than affect. How this connection to a higher power leads to increased social bonding is unclear, however. It could be that connection to a higher power directly leads to further feelings of connectedness, or it could lead to affect changes, which in turn would promote bonding with others. It may be that attachment to God leads to some of the improved wellbeing effects that stem from religiosity [43, 44]. However, Pirutinsky et al. [44] also showed that, in addition to attachment to God, social support significantly predicted the protective effects of religion against mental health problems. This raises the following questions: which aspects of rituals are particularly apt at providing wellbeing effects, and what role a connection to God plays? What of rituals that are not religious?

## Secular rituals

As we progress further into the 21$^{st}$ century, growing numbers of individuals are identifying as non-religious. In 2012, 16% of the world population reported no religious affiliation [45] and these numbers are growing in several countries, such as the UK [46, 47], the USA [48], and other Western countries [49]. Given the positive health effects associated with religious attendance [1], leaving religion or not having a religion, and thus not attending religious rituals, means the positive effects of such rituals can no longer be reaped.

There have, however, been efforts to counteract the loss of religious rituals. For over two centuries there have been attempts at creating secular rituals that mimic religious rituals, such

as Comte's Church of Positivism, established in the 1800s [50]. The Church of Positivism, which had the ambitious aim of becoming the 'Religion of Humanity', held regular services at their own temples in the UK, France and Brazil, though with time they failed to recruit new members and have mostly closed down (the last remaining active church is located in the south of Brazil; for a recent documentary, see [51]).

A more recent, and thus far successful, attempt to develop a secular ritual is that of the Sunday Assembly. This movement was initiated in London, England in 2013 by Sanderson Jones and Pippa Evans, as a way of "doing church without god" [52]. This regular Sunday ritual intentionally mimics Christian Evangelical services with traditional hymns being replaced by popular songs with positive messages, such as U2's '*It's a Beautiful Day*'. Instead of a sermon, a TED-style informative talk is given on various topics, from the importance of local flora on the community's sense of identity to talks on gender equality in modern society. Were one to see a video of a Sunday Assembly taking place without sound, it would be largely indistinguishable from many evangelical churches. Since its inception, the Sunday Assembly has grown to become an international brand of secular ritual, with 22 Assemblies in Europe, another 22 in North America, and five across Australia and New Zealand at the time of writing (see https://sundayassembly.online/find-an-assembly/ for a full list).

Price and Launay [53] recently conducted a longitudinal study with the flagship Sunday Assembly in London to better understand some of the impacts on wellbeing that being part of such a group can provide. Their results suggest that attending the Sunday Assembly services significantly improved a composite measure of wellbeing. In the discussion of their paper, Price and Launay [53] hypothesise that the improved wellbeing they found may stem from the social connections made at Sunday Assembly, not unlike connections found in religion and health research (e.g. [54]).

Some researchers believe that religion provides health benefits in ways that secular means cannot such as through the sense of meaning that religion provides (e.g. [55]). However, Galen [56] has suggested that the link between religiosity/spirituality and factors that lead to wellbeing stems from a congruence fallacy, where religiosity/spirituality is not adequately compared to secular equivalents. In this study we sought to test for the first time whether a secular ritual increased the feelings of social bonding with others in a similar way to a religious ritual. To do so, we recruited participants from Sunday Assembly services, as well as matched Christian churches, and measured levels of social bonding before and after the rituals. Our hypotheses were that: (1) as the Sunday Assembly ritual mimics Sunday church ritual, social bonding will significantly increase at ritual follow-up; (2) The change in social bonding would not be significantly different between participants attending a church service from those attending Sunday Assembly services. As there is research suggesting that emotional state/affect is related to feelings of social connectedness [19, 30–32], (3) we expected that positive changes in affect would predict an increase in social bonding, where increased positive affect and decreased negative affect would lead to an increase in social bonding.

## Method

### Participants

Adult (>18 years) participants were recruited from four Sunday Assemblies in the UK; (a) Central London, (b) Reading, (c) Bristol and (d) London, East End. Four churches from a large dataset [30] were chosen as close matches for total congregation size and level of education.

49 ($M_{age}$ = 39.6, $SD_{age}$ = 12.24) participants were recruited from Sunday Assemblies, of which 16 identified as male, 32 as female and one identified as non-binary, and these were

matched with 50 ($M_{age}$ = 57.8, $SD_{age}$ = 18.08) participants from churches, of which 34 identified as female and the remainder as male. A total of 99 participants ($M_{age}$ = 48.2, $SD_{age}$ = 18.21) were included in the study.

## Materials

**Social bonding.** This scale consists of six questions. Five of these were measured on a 7-point Likert scale: "1, not at all", "2, Very slightly", "3, A little", "4, Moderately", "5, Quite a bit", "6, Very much", and "7, extremely". For the Sunday Assembly participants, these five questions were: (one) "At this moment, how connected do you feel to the people in this Sunday Assembly?" (adapted from [57]); (two) "At this moment, how emotionally close do you feel to the other members of this Sunday Assembly as a whole?"; (three) "Thinking about everyone in this Sunday Assembly now, how much do you trust the others in this group?"; (four) "How much do you like the people in this Sunday Assembly overall?" (adapted from [58]); and (five) "Thinking about everyone in this Sunday Assembly now, do you feel you have a lot in common with others in this congregation?" (adapted from [59]). The last question was the pictorial Inclusions of Others in Self scale (IOS; [60]), which is a series of seven pairs of circles that go from not overlapping (one) to heavily overlapping (seven) to picture how much one's self-identity overlaps with others/the group (see Fig 1). Answers to these questions were averaged into a single social bonding score. For the church populations, we used the same phrasing but references to "this Sunday Assembly" were replaced with "your congregation" e.g. "At this moment, how connected do you feel to the people in your congregation?"

Raykov and Marcoulides [61] and Savalei & Reise [62] suggest conducting a factor analysis on any data that is being used to create an average or summed score to check for which measure of reliability should be used. We conducted such a factor analysis (see analysis script) and found that all items loaded onto a single factor, with mean factor loadings were above 0.7, suggesting that the assumption of essential tau-equivalence was not violated. As such Cronbach's alpha and McDonald's omega should be equal. Here we present only the alpha values, but the analysis script also provides omega values for those sceptical of alpha due to recent critical articles (e.g. [63–65]). Internal reliability for the social bonding measure in the Sunday Assembly participants was α = .93, 95% CI [.90, .96] before the ritual and α = .93, 95% CI [.90, .96] for the post-ritual measure, providing evidence of high scale reliability. For the church participants, these scores were α = .94, 95% CI [.91, .96] and α = .89, 95% CI [.84, .93] for the pre- and post-ritual measures, respectively. The internal reliability for all participants is α = .93,

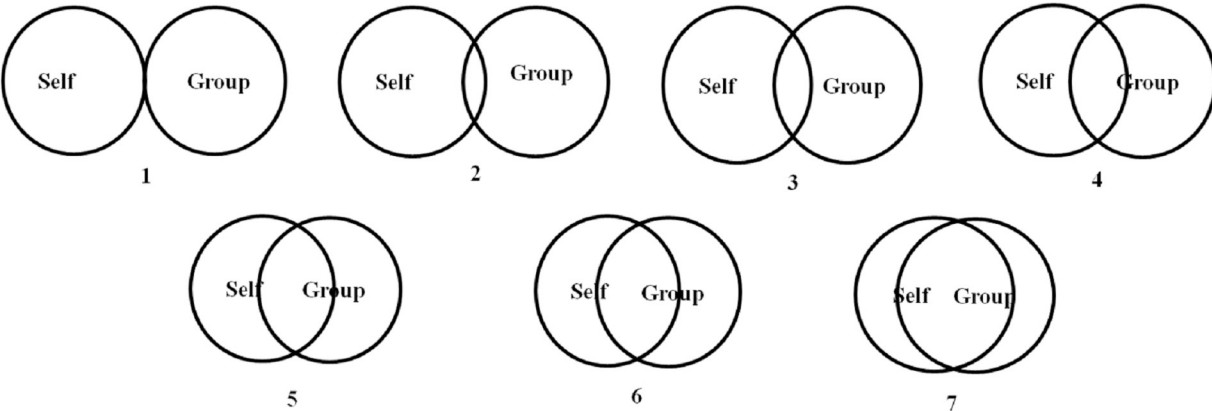

**Fig 1. The inclusion of others in self (adapted from 60) question used as part of the social bonding scale.**

95% CI [.91, .95] and α = .91, 95% CI [.88, .94] for the pre- and post-ritual measures, respectively, also providing evidence of high internal reliability.

**Affect.** We used the Positive And Negative Affect Scale (PANAS; [66]) to collect data on the emotional state of participants both before and after ritual participation. The PANAS asks participants to say how much they are feeling 20 emotions (10 positive, 10 negative) "at this moment" measured on a 6-point Likert scale, from zero to five, where zero is "not at all" and five is "very much". Examples of positive emotions include 'Interested', 'Proud', and 'Inspired', examples of negative emotions include 'Nervous', 'Upset' and 'Irritable'. Watson and colleagues [66] sum the values for the 10 emotions for both positive and negative to provide two scores: the sum of the scores for positive emotions, PANAS+, and the sum of scores for the negative emotions, PANAS–.

**Other measures.** *Religion and Spirituality*. Sunday Assembly participants were asked to rate how religious and spiritual they considered themselves to be, each on a scale of one to seven, where one meant "Not at all" and seven was "Extremely so". Church participants had only been asked to self-rate how religious they considered themselves to be.

Demographic information (age, gender identity, level of education) was recorded.

*Connectedness to God/Something Bigger*. Sunday Assembly participants were asked after their ritual: "During today's Sunday Assembly, did you feel connected to something bigger than yourself, like the universe, and/or feel a sense of awe or wonder?" This was considered analogous to a sense of awe or wonder that may be felt when connected to the divine in a religious ritual, and church participants were asked after their service "During today's service, did you feel connected to God, Jesus, and/or the Holy Spirit?"

**Procedure.** *Assembly recruitment*. Initial contact was made at the flagship London Sunday Assembly. Once the flagship Assembly confirmed their participation, contact was made with other Assemblies. A total of four assemblies, including Central London, agreed to take part.

*Participant recruitment*. Ahead of our attending an Assembly, the Assembly alerted attendees that we would be there via newsletter communications, Facebook posts, and announcements at the previous month's Assembly. On the day of the Assembly, researchers arrived 1 hour before the start of the Assembly. Attendees to the Assembly who arrived anytime between 45 minutes to 5 minutes before the Assembly began were provided with information sheets. Those who opted to take part and met the inclusion criteria were provided with a consent form before taking part.

Participants who took part were provided with a questionnaire, which had an ID code on the front unique to each participant, to ensure anonymity. The pre-Assembly section of the questionnaire contained the PANAS, social bonding questions, and their self-reported level of religiosity and level of spirituality. Halfway through the questionnaire, there was a two-page break which alerted participants that this was the end of the pre-Assembly section. Participants' names were then noted on a post-it note and attached to the questionnaire to match it with the same participant after the Assembly.

After the Assembly, participants returned to the researchers and filled in the post-Assembly half of the questionnaire, which re-measured both PANAS and social bonding and demographic information. The post-it notes were collected and destroyed to ensure anonymity of the data. After this, the data collection was completed. The same procedure was used for church participants.

## Data analysis

A power analysis was conducted to determine the appropriate number of participants. Based on the data reported by Charles et al. [30], the effect size of change in social bonding from

before to after a religious ritual was given by both the conservative $r_R$, [67, 68] and the 'simple difference' effect size $r_K$ [69]. Charles and colleague's [30] UK religious ritual effects had an effect size of $r_R = .34$ and $r_K = .62$, which converts to a $d$ of 0.72 or 1.58, respectively Using G*Power [70], and a one-tailed Wilcoxon-Signed-Rank test (non-parametric, within-participant design) in the $t$-test family with the distribution assumed to be the minimum asymptotic relative efficiency (the most conservative distribution assumption; [71]), $\alpha = 0.05$, Power $(1-\beta) = 0.80$, and an effect size of $d = .72$, we calculated that 16 participants would be needed to have an appropriately powered study for testing the first hypothesis, suggesting this study is appropriately powered.

A similar power analysis for hypothesis two (the comparison between churches and Sunday Assemblies) was also conducted. To detect an interaction effect (before/after ritual interacting with ritual type) using an ANOVA, an effect size of $f = .36$ (converted from $r = .34$), $\alpha = 0.05$, Power $(1-\beta) = 0.80$, 2 groups, 2 measurements, nonsphericity correction of 1 and a correlation among repeated measures of 0.75 (calculated from the data collected in this study), a total of 10 participants would be needed to find a within-factors and/or an interaction effect and 56 total participants to find a between factors effect, suggesting this study is appropriately powered.

## Results

The full analysis script and anonymised dataset have been made available in the supplementary materials.

### Effect of Sunday Assembly attendance on social bonding

A Shapiro-Wilk test was conducted to check whether data met the assumptions for parametric testing. The pre-meeting SB6 score ($W = .977$, $p = .433$) was not significantly different from normal, but the post-meeting SB6 score ($W = .952$, $p = .047$) was. Therefore, non-parametric tests were used.

To test the hypothesis that there would be an increase in self-reported social bonding, from before to after the Sunday Assembly ritual, a one-tailed, non-parametric Wilcoxon signed-rank test was conducted. As predicted, post-Assembly scores ($M = 4.96$, $SD = 1.16$, $Mdn = 5.17$) were significantly higher than at pre-ritual ($M = 4.27$, $SD = 1.26$, $Mdn = 4.33$, $Z = 5.02$, $p < .001$, $r_R = .51$, $r_K = .84$). In this case these results suggest a 'moderate-to-high' effect size according to Fergusson's criteria for social science (Fergusson, 2009). These results remain significant when using a two-tailed test ($p < .001$), and when using the Bonferroni correction for multiple comparisons to account for the follow-up analyses, below ($p < .001$).

### Comparing bonding between church and Sunday Assembly

Both pre- and post-service social bonding scores for church participants were significantly different from normally distributed (W = .927, p = .004 and W = .939, p = .011, respectively). Therefore, a non-parametric form of ANOVA was used for the analysis.

Using the nparLD package of the R coding language, a non-parametric within-between ANOVA was run via the f1.ld.f1 function [72]. The nparLD package's functions provide an ANOVA-like statistic with the denominator degrees of freedom listed as infinite. The f1.ld.f1 function found that there were significant main effects of both time ($F(1, \infty) = 52.06$, $p < .001$) and ritual group ($F(1, \infty) = 4.38$, $p = .036$), but there was no significant interaction effect ($F(1, \infty) = 3.77$, $p = .052$). As seen in Fig 2, there was a significant increase in social bonding from before to after both ritual types, and there was a significantly higher level of social

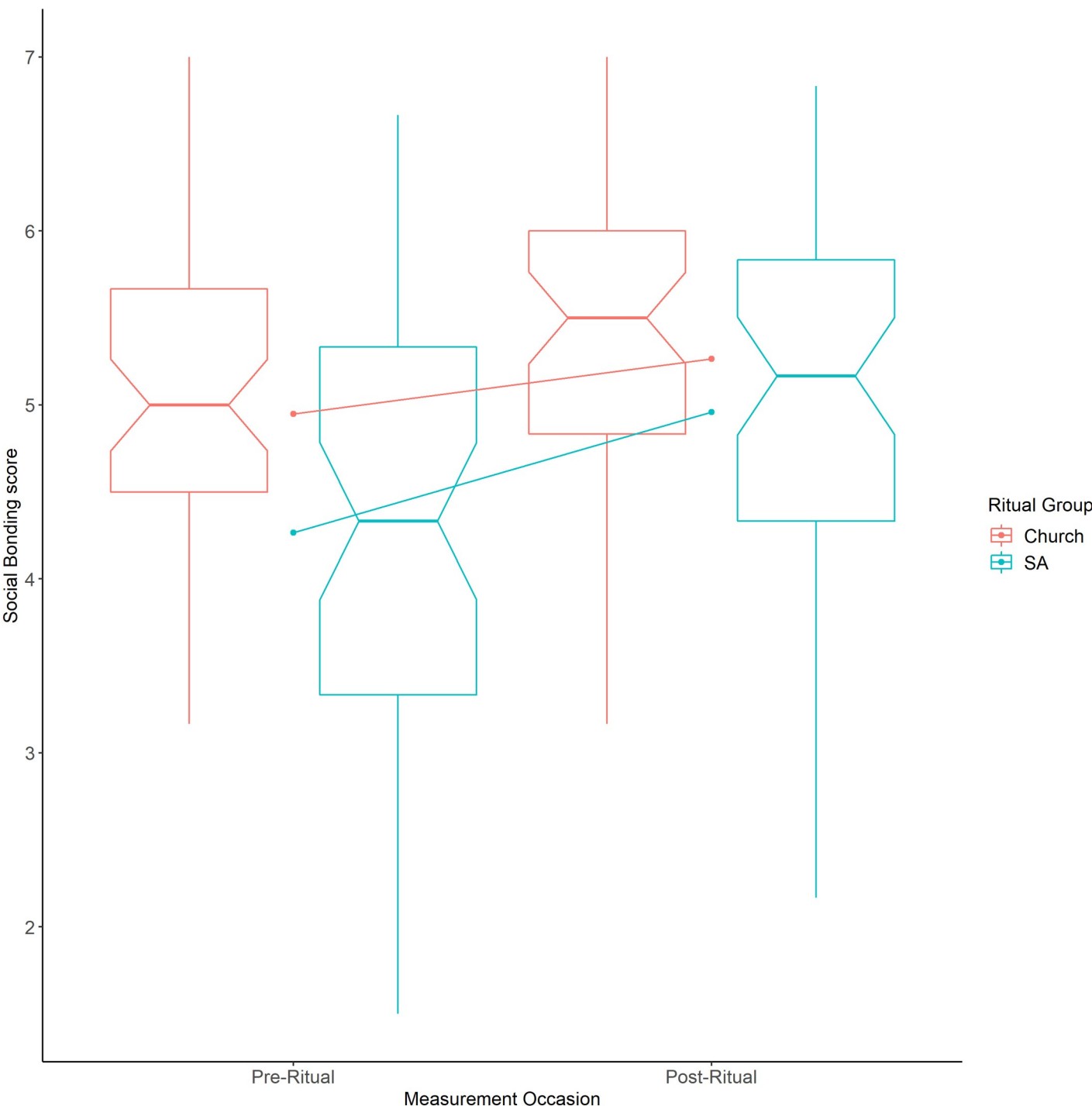

**Fig 2. Notched boxplot showing pre-ritual and post-ritual social bonding in Sunday Assembly (blue) and Church (red) participants.** The notch (indent) around the median shows the 95% confidence interval. The whiskers are +/- 1.5*IQR from the upper and lower quartiles. The diagonal lines show the mean change for the two sets of participants.

bonding at both time points for participants at church than at Sunday Assembly, but there was no significant interaction. In other words, the increase from before to after the ritual was not significantly different between groups.

Note that effect sizes cannot be directly calculated using the non-parametric within-between ANOVA. However, Feys [73] notes that for non-parametric within-between ANOVA tests involving only two time-points (pre-post), there are several alternative ways that effect sizes might be estimated. For completeness, we carried out four alternatives suggested by Feys [73], results of which can be found in the analysis script in the supplemental materials.

We then tested whether affect, as measured by the PANAS, changed from before to after the Sunday Assembly. The pre-meeting PANAS+ ($W$ = .976, $p$ = .445) was not significantly different from normally distributed, however the post-meeting PANAS+ measure ($W$ = .939, $p$ = .017) and both pre- ($W$ = .861, $p < .001$) and post-meeting ($W$ = .738, $p < .001$) PANAS- measures were significantly different from normal. Consequently, non-parametric Wilcoxon signed-ranks tests were conducted.

A two-tailed Wilcoxon signed-ranks test showed that there was a significant change in positive affect, with an increase in positive affect from before ($M$ = 26.21, $SD$ = 9.23, $Mdn$ = 24.0) to after the Sunday Assembly ritual ($M$ = 31.68, $SD$ = 11.57, $Mdn$ = 32.0, $Z$ = 3.90, Bonferroni-corrected $p < .001$, $r_R$ = .40, $r_K$ = .60), as well as a significant decrease in negative affect (PANAS-) from before ($M$ = 5.00, $SD$ = 4.10, $Mdn$ = 4.0) to after the ritual ($M$ = 3.30, $SD$ = 4.23, $Mdn$ = 2.0, $Z$ = 2.41, Bonferroni-corrected $p$ = .047, $r_R$ = .25, $r_K$ = .44). These results suggest that there was a moderate effect of taking part in secular ritual on positive affect and a small effect on negative affect. Fig 3 shows these results and includes the matched church data for comparison.

## The role of affect on social bonding

We then examined whether this change in affect predicted the significant change in social bonding we observed in Sunday Assembly participants. First, we visualised the Sunday Assembly data using a correlation plot (See S1A Fig). To test our hypothesis that affect change predicted social bonding change, we conducted a multiple regression for Sunday Assembly participants with PANAS+ and PANAS- as predictors. Both social bonding and the PANAS subscales were standardised for the regression analysis. A multiple regression showed that there was a significant model ($F(2,44)$ = 16.76, Bonferroni-corrected $p < .001$, $R^2$ = .432, $R_{adj}^2$ = .407), with change in PANAS+ being a significant positive predictor of change in social bonding score, but not PANAS- (see Table 1).

## Exploratory analyses

In their work on religious ritual, Charles, van Mulukom and colleagues [30] found that the feeling of connection to something bigger was also a significant predictor of social bonding. Further, Price and Launay [53] suggested that research should account for the length of time one had been attending Sunday Assembly to determine if this plays a role in some of the effects seen. As such, we conducted an exploratory correlation analysis and created a correlation plot to visualise the analysis. As shown in S1A Fig, both the length of how long one had been attending the Sunday Assembly and the feeling of connection to something bigger were positively correlated with social bonding change. Moreover, baseline social bonding score may play a role in the level of change in bonding, i.e. the lower the starting score, the larger the change. Consequently, we conducted stepwise multiple linear regressions to include the control variable of baseline social bonding score, the level of connection to something bigger and how long (in months) participants had been attending Sunday Assemblies. Control variables of age, education, self-rated spirituality, and self-rated religiosity were also added to the stepwise process to determine if they played a role.

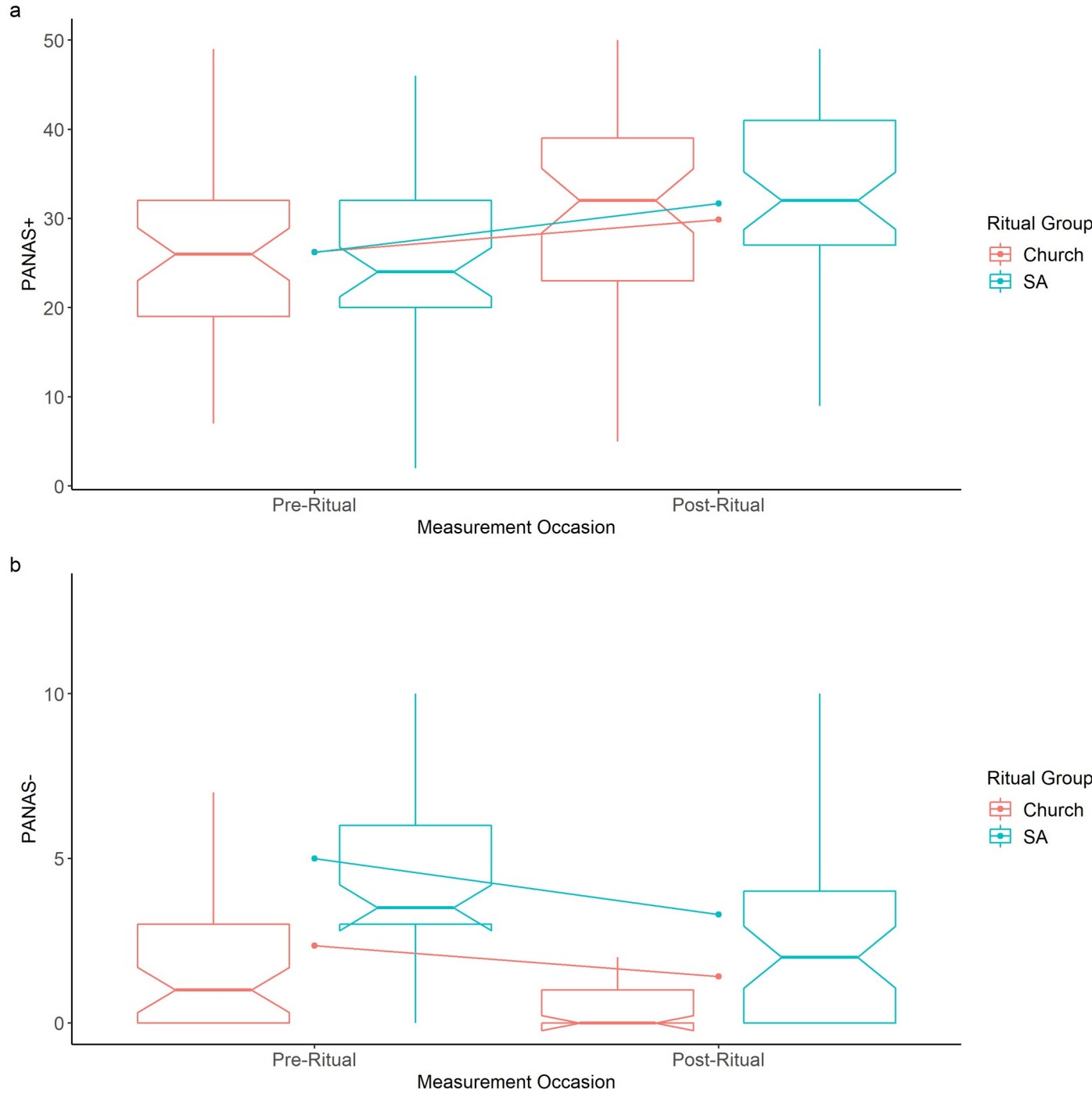

**Fig 3. Notched boxplots showing pre-ritual and post-ritual (a) PANAS+ and (b) PANAS- in Sunday Assembly (blue) and Church (red) participants.** The notch (indent) around the median shows the 95% confidence interval. The whiskers are +/- 1.5*IQR from the upper and lower quartiles. The diagonal lines show the mean change for the two sets of participants.

The stepwise regression analysis was conducted using the stepAIC function in the MASS package in R. This conducts a stepwise regression using the Akaike Information Criterion (AIC; [74]). An AIC provides a numerical indicator of the ratio between the goodness of fit of the model and the simplicity of the model. The lower the AIC value, the lower the level of

**Table 1. Multiple linear regression predicting change in social bonding.**

| Variable | B | 95% CI | β | t | Sig. (p) |
|---|---|---|---|---|---|
| (Constant) | - .00 | [-.23, .23] | | 0.00 | >.999 |
| ΔPANAS+ | .70 | [.45, .95] | .70 | 5.66 | < .001 |
| ΔPANAS- | .14 | [-.11, .39] | .14 | 1.15 | .257 |

*Note*: $R^2$ = .432, 95% CI = [.19, .58]

information loss, meaning it is a better, more parsimonious model (or more likely to be indicative of the true model). The absolute AIC values are not important, but the relative values are compared. The model with the lowest AIC value is the most parsimonious model that best fits the data. This can be run in one of three ways: (one) backwards (aka backwards elimination), starting from a full model with all predictor variables and iteratively removing the predictors that contribute the least to the model; (two) forwards, starting with a null model and adding predictor variables iteratively until the model no longer improves, or (three) both, where a combination of the two methods is used to determine the best model. The both-direction method starts with no predictors and add predictors that contribute the most value (like the forward method) then removes any variables that no longer provide an improvement to model fit (like the backward method). The default method in the stepAIC function is 'both', as this allows predictors to be added and removed at each step, and is best used if conducting exploratory analyses [75, 76].

A both-direction stepwise regression converged on a final model where four variables were included: baseline social bonding, PANAS+ change, connectedness to something bigger and age. The final model was significant ($F(4,42)$ = 23.67, $p < .001$, $R^2$ = .693, $R^2_{Adj}$ = .663), and showed that change in social bonding was significantly predicted by baseline social bonding, PANAS+ change, and connectedness to something bigger. Including age improved the model, but it was not a significant predictor (see Table 2).

## Church exploratory analyses

First data were plotted on a correlation plot to visualise the data (see S1B Fig). As with the Sunday Assembly data, a stepwise regression was conducted on the matching church participants to see if similar results were found in the matched participants. The full model included the same predictors and control variables the Sunday Assembly participants apart from spirituality, as self-rated spirituality was not measured in the survey for church participants. The both-direction stepwise regression converged on a final model where five variables were included, baseline social bonding, PANAS+ change, PANAS- change, connectedness to God, and length

**Table 2. Final model after both-direction stepwise multiple linear regression predicting change in social bonding in Sunday Assembly participants.**

| Variable | B | 95% CI | β | t | Sig. (p) |
|---|---|---|---|---|---|
| (Constant) | - .00 | [-.17, .17] | | 0.00 | >.999 |
| Baseline social bonding | - .48 | [-.65, -.30] | -.48 | -5.47 | < .001 |
| ΔPANAS+ | .50 | [.30, .70] | .50 | 5.09 | < .001 |
| Connected to Something Bigger | .31 | [.11, .51] | .31 | 3.10 | .003 |
| Age | .01 | [-.00, .03] | .16 | 1.88 | .068 |

*Note*: $R^2$ = .693, 95% CI = [.48, .77]

**Table 3. Final model after both-direction stepwise multiple linear regression predicting change in social bonding in church participants.**

| Variable | B | 95% CI | $\beta$ | t | Sig. (p) |
|---|---|---|---|---|---|
| (Constant) | - .00 | [-.23, .23] | | -0.00 | >.999 |
| *Baseline social bonding* | - .42 | [-.69, -.15] | -.42 | -3.13 | .003 |
| *ΔPANAS+* | .40 | [.12, .68] | .40 | 2.85 | .007 |
| *ΔPANAS-* | - .17 | [-.41, .07] | - .17 | - 1.43 | .161 |
| *Connected to Something Bigger* | .27 | [-.02, .56] | .27 | 1.88 | .067 |
| *Months Attended* | .27 | [-.00, .03] | .27 | 2.20 | .033 |

*Note*: $R^2$ = .457, 95% CI = [.16, .57]

of attendance. The final model was significant ($F(5,41) = 6.90$, $p < .001$, $R^2 = .457$, $R^2_{Adj} = .391$), and showed that change in social bonding was significantly predicted by baseline social bonding, PANAS+ change, and length of time one had been attending. Connectedness to God, and PANAS- change improved the model, but were not significant predictors (see Table 3).

## Discussion

Religious rituals occur in all human societies [77], and they seem to confer various benefits to those who take part [1]. It has been suggested that rituals are evolutionarily adaptive by helping foster social bonds [16]. This hypothesis has received some support from field research on religious rituals [30] and from a large body of research showing social bonds provide health benefits [39–41]. It has also been proposed that attending secular rituals, such as Sunday Assembly meetings, may lead to improved wellbeing (e.g. [53]). However, whether the social bonding effect reported from religious rituals is also seen in secular rituals that mimic the behaviours of religious rituals had not been tested before. This study of participants from Sunday Assemblies provides the first evidence that the fostering of social bonds occurs in a secular ritual setting. We compared this to a matched group of individuals from four Christian churches. The results showed that social bonding improvements are of a similar level at both Sunday Assembly and religious ritual settings.

Follow-up analyses found that the increase in social bonding from before to after the Sunday Assemblies was positively predicted by the change in positive affect, as has been found for churches across the UK [30], but not negative affect. These findings are in line with the 'broaden and build' hypothesis, which suggests that positive emotions increase the scope of one's attention to others to allow for the formation of social connections, which themselves lead to improved mental wellbeing [19, 34, 35]. This hypothesis has also been used to suggest that link between religion and wellbeing stems from changes in positive affect [34, 35], which in turn leads to protective social benefits, such as social support [36].

Stepwise regression analysis found that neither level of spirituality nor level of religiosity played a significant role in social bonding change, despite both variables having been related to wellbeing in the past [78]. This could be the result of the methodology of previous studies, which have often used attendance of religious services as a measure for religiosity itself [1, 79], which could conflate the effect of ritual attendance with religiosity and/or spirituality. Diener and colleagues [36] have noted that the reported relationship between religiosity and wellbeing is conditional on social support and social structure. This may explain why religiosity did not directly predict social bonding change in either Sunday Assembly or church participants. Price and Launay [53] have specifically suggested that future research should account for the length of time one had been attending Sunday Assembly, to see if this could explain the wellbeing

effects they reported. In the stepwise regression model, the length of attendance did not add predictive value for the change in social bonding in the Sunday Assembly participants. If, as Price and Launay [53] suggest, the improved wellbeing stems from social bonding, this may suggest that protective effects of participating in secular ritual could occur quickly. We must note, though, that we likely failed to detect this effect for Sunday Assembly participants because there were a number of people attending the ritual for the first time in the Sunday Assembly population, which was not the case with the Christian church participants, for whom we found that length of attendance predicted strength of social bonding. Future research should attempt to account for the effect of newcomers on social bonding during group rituals.

This work is the first to demonstrate that secular rituals, much like religious rituals, promote feelings of social bonding. However, we acknowledge that there are limitations to this study. Firstly, this study was not pre-registered. Given the changes suggested by those promoting Open Science methodologies since the advent of the replication crisis [80–82], the methods and analysis plans could have been registered in advance of conducting the study. Though pre-registration was not done in this case, the full anonymised dataset and the research materials are provided in supplementary materials in accordance with other Open Science practices, and a power analysis was provided to support the sample size used in this study.

Another limitation is that we only conducted research with one type of secular ritual, the Sunday Assembly meetings. Sunday Assembly meetings are not the only secular ritual that mimic religious ritual, with other examples including the Church of Positivism [50]–still active in Brazil–and the Religious Humanism movement in the United States.

One avenue for future research is to conduct studies investigating whether the positive health effects found in those who regularly attend religious rituals can also be seen in those who regularly attend Sunday Assemblies or other similar non-religious rituals that mimic the behaviours of religious rituals, compared to those who do not attend such rituals. Examples of such positive health effects are better immune function and lowering levels of all-cause mortality [3–6], depression [7–9, 83, 84] and suicidality [1]. Here, we have examined the role of ritual on social bonding. However, to better understand the mechanisms underlying the protective factors that have previously been related only to religious participation, future research could compare health outcomes from those who attend secular rituals to those who do not, while taking affect and social bonding into account. We also recommend that, much like in our research, social bonding factors be explicitly measured in future ritual and health research, as this may provide a more comprehensive understanding of the mechanism by which ritual attendance appears to improve wellbeing.

Future research can also look more widely at gatherings of secular groups, which are not intentionally 'rituals' but nonetheless may create a sense of connection to something bigger than oneself. A variety of gatherings may function as a form of 'implicit religion' [85–87], such as sporting events where one feels connected to a team spirit [88, 89], thus creating social bonds in ways similar to religious rituals. Conducting research in such settings would allow us to better understand the nature and effects of ritual-like social bonding in secular contexts.

## Conclusion

Throughout the first two decades of the 21st century, levels of religiosity have been on the decline in many Western countries [45–49]. This may be worrying to some, as there is a wealth of evidence from the religion and health field demonstrating that participation in religious activities provides a variety of health benefits [1]. However, some research suggests that secular rituals, such as the Sunday Assembly, may provide some boost wellbeing [53]. It is thus

possible that the mechanism by which health benefits arise from religious participation are not exclusive to religious rituals. Dunbar [16] provides one possible explanatory mechanism underlying the positive health benefits that seems to stem from rituals: the formation of social bonds. Previous research has demonstrated that religious rituals lead to social bonding [30] and that social bonding leads to better health outcomes [39–41]. In this study we have demonstrated that secular rituals, in the form of Sunday Assembly meetings, also lead to increases in feelings of social bonding, in part via an increase in positive affect. This shows that, for some individuals, secular rituals may serve as an alternative to religious ones.

## Supporting information

**S1 Fig.** Two supplementary figures showing the correlation plots Sunday Assembly participants (A) and one for church participants (B).
(DOCX)

**S1 File. Sunday Assembly questionnaire.**
(DOCX)

**S2 File. Sunday Assembly R script.**
(R)

**S1 Data. Social bonding factor analysis.**
(CSV)

**S2 Data. SA data only–WIDE.**
(CSV)

**S3 Data. Church and SA data–LONG.**
(CSV)

**S4 Data. Church and SA data–WIDE.**
(CSV)

**S5 Data. Church data only–WIDE.**
(CSV)

## Author Contributions

**Conceptualization:** Sarah J. Charles, Valerie van Mulukom, Fraser Watts, Robin I. M. Dunbar, Miguel Farias.

**Data curation:** Sarah J. Charles, Valerie van Mulukom, Jennifer E. Brown.

**Formal analysis:** Sarah J. Charles, Valerie van Mulukom.

**Funding acquisition:** Fraser Watts, Robin I. M. Dunbar, Miguel Farias.

**Investigation:** Sarah J. Charles, Valerie van Mulukom, Jennifer E. Brown, Miguel Farias.

**Methodology:** Sarah J. Charles, Valerie van Mulukom, Fraser Watts, Robin I. M. Dunbar, Miguel Farias.

**Project administration:** Sarah J. Charles, Valerie van Mulukom, Robin I. M. Dunbar, Miguel Farias.

**Supervision:** Valerie van Mulukom, Robin I. M. Dunbar, Miguel Farias.

**Validation:** Valerie van Mulukom.

**Visualization:** Sarah J. Charles.

**Writing – original draft:** Sarah J. Charles, Jennifer E. Brown.

**Writing – review & editing:** Sarah J. Charles, Valerie van Mulukom, Fraser Watts, Robin I. M. Dunbar, Miguel Farias.

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
