## [Decision Letter · Decision Letter 0]

6 Oct 2020

PONE-D-20-19769

United on Sunday: The effects of secular rituals on social bonding and affect

PLOS ONE

Dear Dr. Charles,

Thank you for submitting your manuscript to PLOS ONE. After careful consideration, we feel that it has merit but does not fully meet PLOS ONE’s publication criteria as it currently stands. Therefore, we invite you to submit a revised version of the manuscript that addresses the points raised during the review process.

Please make the minor changes suggested by the reviewers and respond to each comment with the changes made. We look forward to receiving your revised manuscript.

We look forward to receiving your revised manuscript.

Kind regards,

Amy Michelle DeBaets, PhD

Academic Editor

PLOS ONE

Journal Requirements:

2.We noted in your submission details that a portion of your manuscript may have been presented or published elsewhere.

[Yes,

Four churches were chosen from the Charles, van Mulukom et al (2020) paper, currently under review/]

3.Please upload a copy of Figure 3, to which you refer in your text on page 11. If the figure is no longer to be included as part of the submission please remove all reference to it within the text.

Reviewers' comments:

Reviewer's Responses to Questions

**Comments to the Author**

1. Is the manuscript technically sound, and do the data support the conclusions?

Reviewer #1: Yes

Reviewer #2: Yes

2. Has the statistical analysis been performed appropriately and rigorously? 

Reviewer #1: Yes

Reviewer #2: I Don't Know

3. Have the authors made all data underlying the findings in their manuscript fully available?

Reviewer #1: Yes

Reviewer #2: Yes

4. Is the manuscript presented in an intelligible fashion and written in standard English?

Reviewer #1: Yes

Reviewer #2: Yes

5. Review Comments to the Author

Reviewer #1: The study described in this manuscript investigates an important topic that will be of interest to many behavioural science researchers, and seems to have been conducted carefully and competently. The lit review is appropriate and sufficiently thorough, and the statistical analysis and conclusion seem rigorous and reasonable. Therefore I think this manuscript is already in pretty good shape, and in my opinion nearly ready for publication, and so this will be a light-touch review.

Issues that should be addressed:

1. The first reference listed is numbered ‘2’, which is confusing; what happened to reference 1?

2. I am confused by this statement on p. 12: “Both were correlated with bonding change, seen in supplementary figure A.” Apparently this statement refers to an analysis that was done which showed that ‘connection to something bigger’ and ‘length of time’ correlated with bonding change, but it is unclear whose analysis this is. If it is using data from the current study, why is the figure presented before any analysis procedure is actually described? This point of confusion is made worse by the fact that Supplementary Figures A and B are both very low resolution and hard to read, at least in the copy of the manuscript that I received. Please clarify the meaning of this statement, and also include higher-quality image files for these supplementary figures.

3. There are some grammatical problems throughout the manuscript, such as typos and convoluted and confusing sentence structures. Please proofread and correct these as best you can. A few selected examples are below.

P. 2: “Consequently, the Broaden and Build hypothesis linking religious ritual to wellbeing via positive emotions’ role in broadening social bonding (32, 33) is unsurprising”; this sentence is worded in a confusing way, I’ve read it several times and am still not sure I understand it.

P. 3: Please check the meaning of the phrase “begs the question”, as it is used incorrectly here. I think you mean “raises the question”. Also please re-word the sentence that starts with this phrase, as it is confusing; for one thing, it refers to “the question”, but then states not one but three questions in quick succession.

P. 13: “…and showed that in social bonding was…”, please correct.

P. 17: Another confusingly-worded sentence: “However, comparing health outcomes from those who attend secular rituals to those who do not on health effects, while taking affect and social bonding into account may help further understand the mechanisms underlying the protective factors that have previously been related only to religious participation”. Please simplify and re-word so that the meaning is clearer. Also, a word like ‘illuminate’ would be more appropriate here than ‘understand’.

Reviewer #2: This is a very strong paper that presents innovative data on a topic that should be of academic and practical interest to all. Through carefully testing of their hypothesis linking ritual experience to positive affect and social bonding, the authors convincingly argue that secular groups can be an equally powerful setting for receiving the benefits to welfare of religious participation. The methodology is rigorous and clearly explained, and the results raise fascinating questions for future research.

As well as deepening our understanding of the link between ritual and well being, the paper raises a future path for research that drills down into exactly what it is about ritual that produces positive outcomes. To what extent does one have to participate? Are some ritual actions more powerful than others in eliciting social bonding? Will any ritual do?

I have some minor thoughts and queries for the authors.

The first is regarding the use of Christian religious services as a proxy for religious ritual in the control group. In the literature section, I recommend that the authors make clear what kinds of religion have been studied by scholars when theorising the positive benefits of church attendance, and what forms of religion this excludes.

The study rests on a comparison between the Sunday Assembly and Christian Church services, which is understandable given the historical/contextual analogy between the groups and the challenges of the congruence fallacy. However, I would caution against over-generalising the literature in suggesting that religious ritual is equally-powerful in generating social bonding. I wonder, for example, how positive affect and social bonding are generated in atheistic Buddhist communities, or liberal Quaker communities, which commonly hold services without the elements on communal singing and preaching etc. The authors hint at this limitation toward the end of the paper, but a word of caution against generalisation from Christianity to religion/ritual might help guide the reader toward the start of the paper.

Secondly, why were the attendees of the Christian churches asked specifically if they felt connected to God/Jesus/ Holy Spirit, and not asked if they felt connected to something bigger than oneself/ the universe/ a sense of awe, as the participants attending the Sunday Assembly were? By wording the question in this way, the survey appears to preclude those people attending Christian church services who have less doctrine-driven experiences of connection, but nevertheless attend church regularly. In some senses, the question might predetermine a distinction that for some attendees does not exist.

Both of these queries are to say that exactly what makes a ritual efficacious (leading to social bonding and potential health benefits), if it is not religiosity, appears increasingly unclear as a result of this paper. A useful further venture might be to take this study into the realm of implicit religion studies, and consider how the attendance of sporting matches, to give one example, might have similar affects.

Overall, this is an impressively researched, clearly articulated, and engaging peer that is sure to generate excitement from religious scholars around the world.

6. PLOS authors have the option to publish the peer review history of their article (what does this mean?). If published, this will include your full peer review and any attached files.

Reviewer #1: No

Reviewer #2: **Yes: **Hannah R H Gould

---

## [Author Response · Author response to Decision Letter 0]

2 Nov 2020

To Amy Michelle DeBaets, PhD, Academic Editor of PLOS ONE

 We are hereby resubmitting the article “United on Sunday: The effects of secular rituals on social bonding and affect”, which has been amended to address your comments, and those made by the two reviewers. We would first like to say that we thank you and the reviewers for their insightful comments and suggestions. We believe that the article is stronger for having taken on the suggestions and addressing the criticisms raised. 

In your direct comments to us, you asked that we amend the formatting to conform to the PLOS ONE style guide. We have now done this by amending heading font sizes, adding a supplementary information section below the references and updated in-text citations accordingly. Further to your concern of dual publication: The four churches used in this article are a sub-sample from the Charles, van Mulukom et al. (2020) article, which is currently under peer review. They are used as a matched comparison for the data we collected in Sunday Assemblies, and has been approached with a completely different set of analyses and hypotheses. We believe that this way, this does not constitute dual publication. Finally, we would like to apologise for previously having mislabelled figures. We have now updated the figure numbers to be correct and have reuploaded all figures to meet appropriate resolution requirements. 

 Regarding the comments made by Reviewer 1: We have made the changes they have recommended, including clarifying the reasons for conducting correlation plots, reuploading the plots in the supplementary figures S1 file as 300dpi images instead of 70dpi images, and have since gone through the article to reword some other sections that may have previously contained typos, or sections that were more difficult to read. For a full list of how we have addressed the comments, please see the next page. 

 Regarding the comments made by Reviewer 2: Reviewer 2’s comments required more substantial additions to the article, such as an extra paragraph to the discussion regarding implicit religion. These additions have improved the article and have allowed us to ensure that we appropriately describe the scope of our research. Based on their comments, we now emphasise that much of the previous research has been conducted in Western, Christian settings so as to situate our research appropriately and to ensure that we do not claim that the scope of the work we have conducted goes beyond such settings. 

 Thanks again for the feedback on the article and the opportunity to improve the article based on the feedback. 

On behalf of myself and my co-authors,

Yours Sincerely, 

Sarah Charles, MBPsS 

Direct Responses to Reviewer points

Reviewer #1: 

The study described in this manuscript investigates an important topic that will be of interest to many behavioural science researchers, and seems to have been conducted carefully and competently. The lit review is appropriate and sufficiently thorough, and the statistical analysis and conclusion seem rigorous and reasonable. Therefore I think this manuscript is already in pretty good shape, and in my opinion nearly ready for publication, and so this will be a light-touch review.

Issues that should be addressed:

1. The first reference listed is numbered ‘2’, which is confusing; what happened to reference 1?

a. ACTION: We have amended an error being caused by EndNote, where the reference in figure 1 was being listed as reference 1. This has been corrected and the reference list order amended accordingly

2. I am confused by this statement on p. 12: “Both were correlated with bonding change, seen in supplementary figure A.” Apparently this statement refers to an analysis that was done which showed that ‘connection to something bigger’ and ‘length of time’ correlated with bonding change, but it is unclear whose analysis this is. If it is using data from the current study, why is the figure presented before any analysis procedure is actually described? This point of confusion is made worse by the fact that Supplementary Figures A and B are both very low resolution and hard to read, at least in the copy of the manuscript that I received. Please clarify the meaning of this statement, and also include higher-quality image files for these supplementary figures.

a. ACTION: We have clarified that we conducted the correlation analysis and provide a correlation plot of this analysis in the supplementary figure file (in the file named S1 Figures). 

b. ACTION: Attempts have been made to upscale Supplementary Figures A and B. They should now be at 300dpi, instead of R’s default 70dpi. We have upscaled all figures in the article using the same method. 

3. There are some grammatical problems throughout the manuscript, such as typos and convoluted and confusing sentence structures. Please proofread and correct these as best you can. A few selected examples are below.

a. P. 2: “Consequently, the Broaden and Build hypothesis linking religious ritual to wellbeing via positive emotions’ role in broadening social bonding (32, 33) is unsurprising”; this sentence is worded in a confusing way, I’ve read it several times and am still not sure I understand it.

i. ACTION: We have amended the wording of this sentence to help clarify what was meant. It now reads: “Consequently, the Broaden and Build hypothesis linking religious ritual to wellbeing via positive emotions’ role in broadening social bonding (34, 35).”

b. P. 3: Please check the meaning of the phrase “begs the question”, as it is used incorrectly here. I think you mean “raises the question”. Also please re-word the sentence that starts with this phrase, as it is confusing; for one thing, it refers to “the question”, but then states not one but three questions in quick succession.

i. ACTION: We have amended the wording of this sentence to: “This raises the following questions: which aspects of rituals are particularly apt at providing wellbeing effects, and what role a connection to God plays?; what of rituals that are not religious?” We believe this is better phrased and addresses the issue of misusing the term “begs the question”.

c. P. 13: “…and showed that in social bonding was…”, please correct.

i. ACTION: We have moved the tables, which stopped the sentence from running over the page and stopped a premature paragraph break. We also added the missing word “change”, so the sentence now reads: “…and showed that change in social bonding was significantly predicted by baseline social bonding, PANAS+ change, and connectedness to something bigger.”

d. P. 17: Another confusingly-worded sentence: “However, comparing health outcomes from those who attend secular rituals to those who do not on health effects, while taking affect and social bonding into account may help further understand the mechanisms underlying the protective factors that have previously been related only to religious participation”. Please simplify and re-word so that the meaning is clearer. Also, a word like ‘illuminate’ would be more appropriate here than ‘understand’.

i. ACTION: We have reworded this section, so the sentence now reads: “However, to better understand the mechanisms underlying the protective factors that have previously been related only to religious participation, future research could compare health outcomes from those who attend secular rituals to those who do not, while taking affect and social bonding into account.”

Reviewer #2: 

This is a very strong paper that presents innovative data on a topic that should be of academic and practical interest to all. Through carefully testing of their hypothesis linking ritual experience to positive affect and social bonding, the authors convincingly argue that secular groups can be an equally powerful setting for receiving the benefits to welfare of religious participation. The methodology is rigorous and clearly explained, and the results raise fascinating questions for future research.

As well as deepening our understanding of the link between ritual and wellbeing, the paper raises a future path for research that drills down into exactly what it is about ritual that produces positive outcomes. To what extent does one have to participate? Are some ritual actions more powerful than others in eliciting social bonding? Will any ritual do?

I have some minor thoughts and queries for the authors.

1. The first is regarding the use of Christian religious services as a proxy for religious ritual in the control group. In the literature section, I recommend that the authors make clear what kinds of religion have been studied by scholars when theorising the positive benefits of church attendance, and what forms of religion this excludes.

a. ACTION: We fully agree, and thank the reviewer for pointing out this oversight. The majority of studies on health/wellbeing and religion have looked at Western religions. We have noted this in the introduction: “Much of this literature was conducted in western, democratic nations, and within Christian settings, though there are some notable exceptions to this (1-3).”

2. The study rests on a comparison between the Sunday Assembly and Christian Church services, which is understandable given the historical/contextual analogy between the groups and the challenges of the congruence fallacy. However, I would caution against over-generalising the literature in suggesting that religious ritual is equally-powerful in generating social bonding. I wonder, for example, how positive affect and social bonding are generated in atheistic Buddhist communities, or liberal Quaker communities, which commonly hold services without the elements on communal singing and preaching etc. The authors hint at this limitation toward the end of the paper, but a word of caution against generalisation from Christianity to religion/ritual might help guide the reader toward the start of the paper.

a. ACTION: We agree that there may be differences across rituals, even the groups mentioned above have some kind of group activities (whether more meditative or contemplative), which we suggest may create similar social bonding effects. We highlight in the discussion ideas for future research in this area. 

3. Secondly, why were the attendees of the Christian churches asked specifically if they felt connected to God/Jesus/ Holy Spirit, and not asked if they felt connected to something bigger than oneself/ the universe/ a sense of awe, as the participants attending the Sunday Assembly were? By wording the question in this way, the survey appears to preclude those people attending Christian church services who have less doctrine-driven experiences of connection, but nevertheless attend church regularly. In some senses, the question might predetermine a distinction that for some attendees does not exist.

a. RESPONSE: We understand where the reviewer is coming from and we agree that many people sensing a connection to God will experience this with awe and as a connection to something bigger than themselves (e.g. the literature on religion and spirituality shows that for many religious people these two experiences overlap and are not separate). However, we argue that the goal of religious ritual is largely focused on a connection with a specific higher power (e.g. 4), and therefore we adapted the specific survey question phrasing to more exactly tap into the religious or secular context. 

4. Both of these queries are to say that exactly what makes a ritual efficacious (leading to social bonding and potential health benefits), if it is not religiosity, appears increasingly unclear as a result of this paper. A useful further venture might be to take this study into the realm of implicit religion studies, and consider how the attendance of sporting matches, to give one example, might have similar affects.

a. ACTION: Thank you for this insight. We have added, in the discussion section, suggestions of further research that include implicit religion: “Future research can also look more widely at gatherings of secular groups, which are not intentionally ‘rituals’ but nonetheless may create a sense of connection to something bigger than oneself. A variety of gatherings may function as a form of ‘implicit religion’ (5-7), such as sporting events where one feels connected to a team spirit (8, 9), thus creating social bonds in ways similar to religious rituals. Conducting research in such settings would allow us to better understand the nature and effects of ritual-like social bonding in secular contexts.”

References

1. Loewenthal KM, Dein S. Religious Ritual and Wellbeing. Applied Jewish values in social sciences and psychology: Springer; 2016. p. 151-63.

2. Chang W-C. Religious attendance and subjective well-being in an Eastern-culture country: Empirical evidence from Taiwan. Marburg Journal of Religion. 2009;14(1).

3. Roemer MK. Religion and subjective well-being in Japan. Review of Religious Research. 2010:411-27.

4. Van Cappellen P. Rethinking self-transcendent positive emotions and religion: Insights from psychological and biblical research. Psychology of Religion and Spirituality. 2017;9(3):254.

5. Aicinena S. Implicit religion and the use of prayer in sport. American Journal of Sociological Research. 2017;7(1):56-65.

6. Lord K. Implicit Religion: Definition and Application. Implicit Religion. 2006;9(2).

7. Bailey E. Introduction. The Notion of Implicit Religion: What it Means, and Does Not Mean. The Secular Quest for Meaning in Life Denton Papers in Implicit Religions. 2002:1-11.

8. Sullivan GB. Collective emotions: A case study of South African pride, euphoria and unity in the context of the 2010 FIFA World Cup. Frontiers in psychology. 2018;9:1252.

9. Halldorsson V. National sport success and the emergent social atmosphere: The case of Iceland. International Review for the Sociology of Sport. 2020:1012690220912415.

---

## [Editor Report · Decision Letter 1]

5 Nov 2020

United on Sunday: The effects of secular rituals on social bonding and affect

PONE-D-20-19769R1

Dear Dr. Charles,

We’re pleased to inform you that your manuscript has been judged scientifically suitable for publication and will be formally accepted for publication once it meets all outstanding technical requirements.

Kind regards,

Amy Michelle DeBaets, PhD

Academic Editor

PLOS ONE
---

## [Editor Report · Acceptance letter]

17 Nov 2020

PONE-D-20-19769R1 

United on Sunday: The effects of secular rituals on social bonding and affect 

Dear Dr. Charles:

I'm pleased to inform you that your manuscript has been deemed suitable for publication in PLOS ONE. Congratulations! Your manuscript is now with our production department. 

Kind regards, 

on behalf of

Dr. Amy Michelle DeBaets 

Academic Editor

PLOS ONE